# How to deal w\_\_\_ missing input data

Martin Gauch<sup>1</sup>, Frederik Kratzert<sup>2</sup>, Daniel Klotz<sup>2,3</sup>, Grey Nearing<sup>1</sup>, Deborah Cohen<sup>4</sup>, and Oren Gilon<sup>4</sup>

<sup>1</sup>Google Research, Zurich, Switzerland

<sup>2</sup>Google Research, Vienna, Austria

<sup>3</sup>IT:U Interdisciplinary Transformation University, Linz, Austria

<sup>4</sup>Google Research, Tel Aviv, Israel

**Correspondence:** Martin Gauch (gauch@google.com)

Abstract. Deep learning hydrologic models have made their way from research to applications. More and more national hydrometeorological agencies, hydro power operators, and engineering consulting companies are building Long Short-Term Memory (LSTM) models for operational use cases. All of these efforts come across similar sets of challenges—challenges that are different from those in controlled scientific studies. In this paper, we tackle one of these issues: how to deal with missing input data? Operational systems depend on the real-time availability of various data products—most notably, meteorological forcings. The more external dependencies a model has, however, the more likely it is to experience an outage in one of them. We introduce and compare three different solutions that can generate predictions even when some of the meteorological input data do not arrive in time, or not arrive at all—: First, *input replacing*, which imputes missing values with a fixed number; second, *masked mean*, which averages embeddings of the forcings that are available at a given time step; third, *attention*, a generalization of the masked mean mechanism that dynamically weights the embeddings. We compare the approaches in different missing data scenarios and find that, by a small margin, the masked mean approach tends to perform best.

# 1 Introduction

Deep learning approaches for hydrologic modeling are now making their way from research settings into real-world operational deployments (e.g., Nearing et al., 2024; Frame et al., 2025; Read et al., 2021; Franken et al., 2022). Unfortunately, the real world is messy and in many ways does not conform to the controlled settings we can assume in research studies (Mitchell and Jolley, 1988). One prime example for such complications is the occurrence of outages with input data products: state-of-the-art operational hydrologic models rely on the real-time availability of several externally provided meteorological forcing products. As an example, the hydrologic model in Google's flood forecasting system uses four different weather data products from four different data providers as inputs (Cohen, 2024). At any point in time, one or more of these providers might experience an outage and not deliver the data in time to make the next prediction. Where the timely arrival of data is usually not an issue in research contexts, not producing forecasts for days or even weeks is not an option for operational systems that are needed for flood forecasts or water management.

Moreover, models that can cope with missing input data are useful in other settings, such as training on data products that are available for different time periods or different spatial extents: the observation that larger and more diverse training sets

**Figure 1.** Different scenarios for missing input data (gray bars): outages at individual time steps (top), data products starting at different points in time (middle), and local data products that are not available for all basins (bottom). All of these scenarios reduce the number of training samples for models that are not robust, i.e., that cannot cope with missing data (yellow, small box), while the models presented in this paper are robust, i.e., they can be trained on all samples with valid targets (purple, large box).

- generally benefit the prediction quality (Kratzert et al., 2024) appears at odds with the fact that local meteorological forcings tend to have higher resolution and be more accurate than global ones (Clerc-Schwarzenbach et al., 2024). Our proposed methods can mitigate this tension, as they allow us to train a single global model that incorporates local forcings where they are available (Fig. 1). Orthogonally to spatial coverage, our methods further allow us to train models with forcings that have different temporal coverage. This is especially useful for more recent data products based on remote sensing information.
- Inevitably, the quality of predictions degrades as fewer input data products are available (Kratzert et al., 2021). Fortunately, deep learning methods are flexible enough to offer solutions that limit this decay while remaining competitively accurate when all data are available. In the following sections, we present three strategies to accomplish this goal:
  - First, **input replacing** replaces missing forcing data with a fixed value and adds binary flags to indicate outages.
  - Second, masked mean embeds each forcing product separately and averages the embeddings of all products that are available at a given time.
  - Third, we show how the masked mean strategy is a special case of a theoretically more expressive but practically equally accurate attention mechanism (Bahdanau et al., 2015) that can dynamically adjust the weighting of each forcing product, e.g., depending on the static attributes of a basin.

We evaluate these strategies in three settings:

35

- First, random **time step dropout**. We investigate how accuracy deteriorates as forcings are missing at more and more time steps during training and inference -(corresponding to the top row in Fig. 1).

- Second, sequence dropout. We investigate how accuracy deteriorates as certain forcings become entirely unavailable during inference -(corresponding to the middle row in Fig. 1).
- Third, **regional forcing products**. We investigate how the proposed strategies allow training global models that leverage regional forcing data -(corresponding to the bottom row in Fig. 1).

We are not the first to study deep learning models that are robust to missing input data (Afifi and Elashoff, 1966). In fact, today's large language models rely heavily on learning schemes that train the model to predict words given incomplete and masked-out input sentences (e.g., Devlin et al., 2019; Raffel et al., 2020; Brown et al., 2020). These masked language models use special mask tokens to indicate dropped-out data, which—at a high level—are similar to the binary indicators we use in the input replacing strategy. Similar techniques are used in computer vision models, such as Masked Autoencoders (He et al., 2022). Srivastava et al. (2014) highlight an additional benefit of dropping out inputs (or hidden activations) during training: dropout has a regularizing effect on training and therefore reduces overfitting and leads to models that generalize better.

Data-driven methods are also used to explicitly impute missing data (e.g., Schafer, 1997; Wu et al., 2020), including in hydrological and meteorological applications (e.g., Gao et al., 2018; Yozgatligil et al., 2013). Imputation subsequently allows using downstream models that cannot cope with missing data. However, this strategy requires an additional imputation model that needs to be trained separately or jointly with the downstream model, making the setup and training more complex. As we are less focused on the reconstruction of missing data and more focused on maintaining prediction accuracy, we do not consider such approaches in this study.

Our masked mean and attention mechanisms also bear similarity to deep learning approaches that merge multi-modal input data, such as LANISTR (Ebrahimi et al., 2023). Their approach merges inputs from different modalities (such as images, text, or structured data) into a joint embedding space, while allowing individual modalities to be missing at training or inference time. Further, the attention mechanism's dynamic weighting of forcing embeddings can be seen as a variant of the conditioning operation described by Perez et al. (2018) and at a higher level by Dumoulin et al. (2018).

### 2 Data and methods

### 65 **2.1** Data

45

We ran all experiments on the 531 basins of the CAMELS dataset (Newman et al., 2015; Addor et al., 2017) that previous studies used, e.g., Kratzert et al. (2021). The CAMELS dataset comes with three sets of daily meteorological forcings: Daymet (Thornton et al., 1997), Maurer (Maurer et al., 2002), and NLDAS (Xia et al., 2012). We consider these forcings the "external dependencies" in this study. All models use all 15 forcing variables (precipitation, solar radiation, min/max temperature, and vapor pressure for each of the three forcing products) and the same set of 26 static attributes as Kratzert et al. (2021).<sup>1</sup> All models are trained with streamflow as the target variable.

<sup>&</sup>lt;sup>1</sup>Unlike what is mentioned in the paperKratzert et al. (2021), p\_seasonality was actually not used as a static input, as the experiment configuration files show.

**Figure 2.** Illustration of the input replacing strategy. Each box represents an input variable (like precipitation, temperature) from one of the forcing groups. NaNs in the input data for a given time step are replaced by zeros (gray boxes for forcing group 2), all forcings are concatenated, together with one binary flag for each forcing group which indicates whether that group was NaN or not. The resulting vector is passed through an embedding network to the LSTM.

Again following Kratzert et al. (2021), we trained our models on the period 1 October 1990-1999 to 30 September 2008, validated on 1 October 1980 to 30 September 1989, and tested them on 1 October 1989 to 30 September 1999. All results in this paper refer to the test period.

#### **75 2.2 Methods**

The models we train in this paper closely follow the architecture that was used in Kratzert et al. (2021), except that we employ different mechanisms to feed the input data into the LSTM itself. The following paragraphs describe these approaches in more detail.

# 2.2.1 Input replacing

The first mechanism to cope with missing input data sets any missing values to a fixed value and adds a binary flag to indicate these replacements, before concatenating all input data and flags (Fig. 2; see also Nearing et al., 2024). Optionally, we can embed the concatenated vector (in our case, through a small fully-connected network). This reduces the feature dimensions before the vector is finally used as input to the LSTM. Further, we can concatenate a positional encoding vector to the forcings before the embedding, making the model aware of the current input's position relative to the overall sequence length (not shown in Fig. 2). In initial experiments, we also tried to make the replacement value a learned parameter instead of setting it to a fixed value, but we did not see meaningful improvements when doing so. Hence, in all subsequent experiments, we used zero as the fixed value.

## 2.2.2 Masked mean

This approach embeds the forcings of each provider through individual embedding networks, each of them yielding an embedding vector of the same size. At every input time step, we average the non-NaN embeddings of that time step (i.e., the embeddings that correspond to providers that were available at that time step; hence the name "masked mean") and pass the

**Figure 3.** Illustration of the masked mean strategy. Each forcing provider is projected to the same size through its own embedding network. The resulting embeddings of valid providers are averaged and passed on to the LSTM.

resulting joint embedding on to the LSTM (Fig. 3). The inputs to the embedding networks could be extended by additional features, such as the static catchment attributes. However, in our experiments we found that this deteriorated the performance. The flood forecasting system described by Cohen (2024) uses a masked mean approach in the current operational model.

### 95 **2.2.3** Attention

105

Readers who are familiar with deep learning might recognize the masked mean architecture as the simplification of a more general attention mechanism (Bahdanau et al., 2015). Attention mechanisms have become ubiquitous in deep learning, as they are the core component of the popular Transformer architecture (Vaswani et al., 2017). The most common realizations of attention allow the model to dynamically adjust its focus on different input time steps. Appendix D provides a brief introduction to the concept of attention for readers who are not familiar with the topic.

In our case, we apply the attention mechanism over the different available providers at each time step. Figure 4 illustrates the process. Similar to the masked mean approach, we embed each forcing with its own embedding network, resulting in vectors that we use as both the keys and values of the attention mechanism. Additionally, we concatenate the static attributes with a positional embedding of the input time step (not shown in Fig. 4 for brevity) and three binary flags that indicate the availability of each forcing product at the given time step. A separate embedding of the resulting concatenated vector acts as the query. Based on the similarity of the query and each of the key vectors, we obtain a weighting by which we average the values, i.e., the embedding vectors of the forcing products. This weighted average is the input to the LSTM. Hence, the attention mechanism, could—at least in theory—learn to dynamically adjust its focus on each forcing product based on the basin it is asked to predict.

**Figure 4.** Illustration of the attention embedding strategy. Each forcing provider is projected to the same size through its own embedding network. The resulting embedding vectors become the keys (k) and values (v). The static attributes, together with a binary flag for each provider, serve as the query. The attention-weighted average of embeddings is passed on to the LSTM.

# 110 2.3 Experiments

120

We conducted three experiments to test how well each architecture can cope with different scenarios where input data are missing in certain temporal periods or spatial regions. To save computational resources, we performed one hyperparameter tuning and used the resulting best hyperparameters for all further experiments. Appendix A covers our tuning procedure in more detail. In all experiments, we trained each model with three different random seeds.

### 115 2.3.1 Experiment 1: Forcings missing at individual time steps

This experiment simulates short-term outages of certain input products. Because the LSTMs used in hydrologic applications typically ingest input data with one year (365 days) of lookback, even an outage for a single time step can cause problems for the next year to come: for the next 365 days, there will be a NaN input time step, which breaks models that cannot deal with missing input data. We trained and evaluated the different models with an increasing probability of randomly missing input time steps. The time step dropout is sampled independently at random, i.e., at each input time step, each forcing is missing with probability  $p_{\text{time}}$ . This means that all products can be missing at once for certain time steps. We sweep  $p_{\text{time}}$  from 0 to 0.6 in increments of 0.1.

As baselines, we used the three-forcing model from Kratzert et al. (2021). This shows the upper bound of performance we can expect when no data are missing. We also included the worst of the three single-forcing models (based solely on NLDAS) from the same source as a point of reference.

# 2.3.2 Experiment 2: Forcings missing for the entire time sequence

This experiment simulates extended time periods with missing input data. In practical applications, this may happen when an input product has limited temporal coverage, either because it became available later than other products, or because it went out of service or had an extended outage while the model was still in use. We evaluated this scenario by running inference with samples where all time steps of one or two providers were set to NaN, and we report the results for each combination of one or two missing providers.

To make sure the models can cope with this scenario, we trained the models with samples that contained NaNs of two types: (1) dropout of individual time steps (as in the previous experiment) with  $p_{\text{time}} = 0.1$ , and (2) dropout of entire input sequences with  $p_{\text{sequence}} = 0.1$ . We made sure to never drop all three sequences entirely, but allowed the case where all three products are missing at individual time steps.

The natural baselines in these experiments are the corresponding one- and two-forcing models from Kratzert et al. (2021). These baselines are not robust to missing input data, and they simply ingest the concatenated forcing variables from one or two forcing groups.

# 2.3.3 Experiment 3: Forcings missing for certain spatial regions

- Finally, we explored how the different approaches to missing input data fare in settings where an input product is missing for certain regions in space. This is relevant because for many regions there exist local meteorological data products that are of higher quality than globally available ones. At the same time, training on diverse sets of basins benefits performance (see Kratzert et al., 2024). Hence, being able to merge local high-quality forcing data with global streamflow could—at least in theory—combine the best of two worlds.
- We simulated this setting on the CAMELS dataset by training models that received Daymet and Maurer forcings everywhere, but NLDAS forcings only for the 51 basins in the Ohio, Cumberland, and Tennessee River basins (USGS site numbers starting in 03, cf. Wells, 1960, depicted in Fig. 5). As baselines, we trained a model on all three forcings but only the 51 basins, and a model on all 531 basins but only the two forcings that we assumed as available anywhere (Daymet and Maurer).

## 3 Results

150

135

#### 3.1 Experiment 1: Forcings missing at individual time steps

In the first experiment, we trained models at different probabilities  $p_{\text{time}}$  of input products being NaN at individual time steps. Figure 6 shows the resulting Nash–Sutcliffe efficiency (NSE; Nash and Sutcliffe, 1970) and Kling–Gupta efficiency (KGE; Gupta et al., 2009) values at  $p_{\text{time}} = 0.0, 0.1, \dots, 0.6$ , and Appendix C contains plots with additional metrics. As expected, the accuracy of all methods drops with increasing amounts of NaNs. At 0% NaNs, all methods perform roughly as good as the

<sup>&</sup>lt;sup>2</sup>We also performed some preliminary experiments with  $p_{\text{time}} = 0.0$ ,  $p_{\text{sequence}} = 0.1$  since this more closely matches the evaluation setup, but saw no meaningful differences in the results.

**Figure 5.** Map of the 531 CAMELS basins used in this study, with For the 51 basins in the Ohio, Cumberland, and Tennessee River basins highlighted in (purple), we assumed all three forcing to be available. For all other basins (blue), we assumed only Daymet and Maurer forcings to be available.

three-forcing baseline from Kratzert et al. (2021), which cannot cope with missing input data. The models exhibit slightly worse NSE values than the baseline, while masked mean is and input replacing are slightly better in KGE. These minor differences arise because our newly trained models were tuned for a setting with moderate amounts of missing input data and therefore use slightly different hyperparameters than the three-forcings baseline.

As  $p_{\text{time}}$  increases, we see no clear winner in terms of NSE; all methods decay by roughly equal amounts in this metric. For KGE, the masked mean architecture tends to perform better than input replacing and attention: except for  $p_{\text{time}} = 0.2$ , the masked mean results are significantly better than those of input replacing (one-sided Wilcoxon signed-rank test at  $\alpha = 0.05$ ). The attention mechanism generally performs significantly worse than masked mean and input replacing, except at the highest missing data probabilities. To investigate the reason for this why attention under-performs at low  $p_{\text{time}}$ , we plotted the attention weights placed by the model on each set of forcings (Fig. C3), and found that, apart from a select few basins, the weights fluctuate closely around 1/3. Hence, the model merely learned attempted to recover the solution that is hard-coded in the masked mean strategy.

### 3.2 Experiment 2: Forcings missing for the entire time sequence

160

165

In this experiment, we evaluated to what extent the different architectures can maintain their accuracy when one or two sets of forcings are missing entirely at inference time. Figure 7 shows the resulting empirical cumulative distribution functions (CDFs) of NSE values. Kratzert et al. (2021) already provide results which indicate that the availability of fewer forcing products implies worse model performance.

**Figure 6.** Median NSE and KGE across 531 basins at different amounts of missing input time steps. The dotted horizontal line provides the baseline of a model that cannot deal with missing data but is trained to ingest all three forcing groups at every time step. The dashed line represents the baseline of a model that uses the worst individual set of forcings (NLDAS). Both baselines stem from Kratzert et al. (2021). The shaded areas indicate the spread between minimum and maximum values across three seeds; the solid lines represent the median.

The results from experiment 2 corroborate this finding. Except for the NLDAS-only evaluation the experiments where one set of forcings is available at inference time (first column in Fig. 7), the baseline trained on the exact forcings that are available at inference time that one set of forcings (dashed line) performs significantly better than the missing-inputs architectures, but the effect sizes in the comparison to masked mean and attention are small (Cohen's d 

**Figure 7.** Empirical cumulative distribution functions of NSE values across all 531 basins when two (first column) or one (second column) forcing groups are continuously missing. The subplot titles denote which products we passed to the model ingestedduring inference. The dotted line represents the performance of upper bound baseline, a model that is trained and evaluated with all three forcings available; the dashed line represents the performance of a model trained specifically for the available combination of inputsforcings. All results show the mean performance across three seeds; curves further to the right are better.

signed-rank test,  $\alpha = 0.05$ ). This pattern is similar for the additional metrics from Appendix C. However, from a practical hydrological perspective, all approaches perform quite similar, despite the statistical significance.

**Figure 8.** Empirical CDFs of NSE values across the 51 basins of the Ohio, Cumberland, and Tennessee River basins. The dashed line represents the baseline model trained only on those basins but with all forcings. The dotted line is the baseline two-forcing model trained on all 531 basins. The other models are trained on all 531 basins with NLDAS set to NaN outside of the 51 basins. All results show the mean performance across three seeds.

## 190 4 Discussion and conclusions

In this study, we presented three different strategies to build models that can provide streamflow predictions when parts of the meteorological input data are missing. *Input replacing* replaces NaNs with a fixed value, concatenates all forcings, and adds binary flags to indicate the missing data. *Masked mean* embeds each forcing product separately and averages the embeddings of available forcings. Finally, *attention* generalizes the masked mean approach and dynamically calculates a weighting of the different embeddings. Across all experiments (missing individual time steps, missing sequences, regional forcings), the masked mean strategy tends to perform best, although the differences are often small and depend on metrics. The fact that the models are unable to outperform the baseline trained on all three forcings but only 51 local basins (experiment 3) lets us conjecture that the high-quality CAMELS forcings may not be the ideal testbed for an evaluation of regional forcings. All three forcings are of similar quality and the basins in the chosen region are comparably similar and easy to predict, hence, a rather small set of training gauges appears to already yield satisfactory predictions and it becomes difficult to discern meaningful differences. We therefore hypothesize that evaluations on larger datasets and with forcings of more varied quality would yield clearer conclusions. Unfortunately, these larger datasets are still missing the type of widely accepted baseline models and state-of-the-art LSTM configurations that exist for CAMELS. Hence, for this study we chose to stick with the CAMELS dataset in order to maintain consistency with Kratzert et al. (2021) and to allow for easy reproduction of experiments with limited resources. We see great potential for future work that extends the experiment to such settings.

Notably, the attention mechanism—despite being strictly more expressive than the masked mean strategy—does not improve upon these results and largely learns to recover the masked mean solution. We also experimented with analyzing the attention

weights grouped by time steps with falling/rising streamflow or by the forcing whose precipitation deviated the furthest from the mean, but could not identify any patterns (results not shown). Therefore, in its current form, attention appears unnecessary. Nevertheless, we do encourage further work in this direction as our experiments do not fully exhaust the space of possible attention configurations, and we hypothesize that attention might play to its strengths especially in settings where the quality of inputs varies significantly across forcings, space, or time. Extending the scope beyond established baselines, future work could evaluate this, for example, with the new Caravan MultiMet dataset (Shalev and Kratzert, 2024). Caravan MultiMet provides forcings from seven different providers for all basins in the Caravan dataset and its extensions (Kratzert et al., 2023). There are also many alternative approaches to calculating query, keys, and values: e.g., incorporating the forcing information also into the query vector or incorporating static information into the keys and values.

Lastly, we would like to look at the presented strategies from a different perspective: we can view them as means to *inject* additional data into a model. Such injections can happen already during training (the multiple forcings we use in our experiments are an example for this), but they could also happen after training: for example, hydromet agencies could download a publicly available global model and inject locally available forcings or even lagged observations into the model. We encourage exploring such approaches further, as they could alleviate current trade-offs between training set size and input data resolution.

Code and data availability. We conducted all experiments with the NeuralHydrology library (Kratzert et al., 2022). The CAMELS dataset necessary to run the experiments is available at https://ral.ucar.edu/solutions/products/camels (Newman et al., 2015; Addor et al., 2017). The extended Maurer and NLDAS forcings (which include daily minimum and maximum temperature) are available at https://doi.org/10.4211/hs.0a68bfd7ddf642a8be9041d60f40868c. The additional code for analyses and figures presented in this paper are available at https://github.com/gauchm/missing-inputs. Finally, all trained models and results files are available at https://doi.org/10.5281/zenodo.15008460.

### Appendix A: Hyperparameter tuning

All hyperparameter tuning experiments used  $p_{\text{time}} = p_{\text{sequence}} = 0.1$ . We chose these values as an intermediate level of missing data to avoid the computational expense of tuning each architecture for each experiment setup separately. As we built upon the established baselines from Kratzert et al. (2021), we did not tune the LSTM architecture itself for the experiments in this paper. Hence, all LSTMs are trained with 365 daily input time steps, a hidden size of 256, batch size 256, dropout fraction of 0.4 on the output head, and an Adam optimizer with initial learning rate of 1e - 3, which we lowered to 5e - 4 in epoch 10 and to 1e - 4 in epoch 25. We used the NSE\* loss function from Kratzert et al. (2019). For a more in-depth description of these settings, we refer to Kratzert et al. (2019).

We did, however, tune the hyperparameters of the missing-inputs mechanisms as well as the number of training epochs. For input replacing configurations, we chose slightly larger embedding sizes, such that the total parameter count in input replacing configurations is roughly equal to the parameter count in masked mean configurations. Attention configurations are marginally

**Table A1.** Hyperparameter tuning grid.

| Hyperparameter                                |                                          | Values                                                                                                                                                 |  |
|-----------------------------------------------|------------------------------------------|--------------------------------------------------------------------------------------------------------------------------------------------------------|--|
| Embedding hidden layer sizes (ReLU-activated) | Input replacing  Masked mean  Attention* | [5], [7, 5], [17, 10], [17, 17, 10], [17, 17, 17, 10]<br>[5], [5, 5], [10, 10], [10, 10, 10], [10, 10, 10, 10]<br>[10, 10], [10, 10, 10], [10, 10, 10] |  |
| Positional encoding size                      |                                          | 0, 5                                                                                                                                                   |  |
| Number of attention heads                     |                                          | 1, 2, 5                                                                                                                                                |  |
| Evaluated epochs                              |                                          | 5, 10, 15, 20, 25, 30, 35, 40                                                                                                                          |  |

<sup>\*</sup> We excluded configurations with hidden size 5, because the final embedding size must be divisible by the number of attention heads.

**Table A2.** Best hyperparameter configurations based on validation period results.

| Architecture    | Embedding hidden | Positional    | Number of       | Epoch |
|-----------------|------------------|---------------|-----------------|-------|
|                 | layer sizes      | encoding size | attention heads |       |
| Input replacing | [17, 10]         | 5             | _               | 30    |
| Masked mean     | [10, 10, 10, 10] | 0             | _               | 35    |
| Attention       | [10, 10, 10]     | 5             | 1               | 30    |

larger as they have an additional query embedding network, but we consider this difference irrelevant for the results in our comparisons—especially given that the optimal attention configuration was not the largest one in the hyperparameter grid.

We performed a grid search of the hyperparameter combinations listed in Table A1. As for the main experiments, we trained each combination with three different random seeds. Finally, we chose the best configuration for each architecture as the one with the best median NSE value across all basins in the validation period, averaged across seeds. Table A2 lists the best configuration for each architecture.

#### 245 Appendix B: Computational resources

250

We conducted all experiments on Nvidia P100 GPU machines running Python 3.11 and NeuralHydrology 1.11.0 (with local modifications that are part of the 1.12.0 release). In total, including preliminary experiments, hyperparameter tuning, and final experiments, we trained approximately 800 models. This amounts to approximately 286 wall-time computation days (measuring the time from writing the configuration to disk to the last Tensorboard update). We did not spend any effort optimizing the runtime of these jobs; many runs could have been sped up significantly, e.g., through increased parallelism in data loading.

# Appendix C: Additional figures

255

In consideration of the fact that no single metric adequately captures the quality of a model (Gauch et al., 2023), we provide Fig. C1 as an extended version of Fig. 6 (showing the performance with increasing number of NaN inputs for a variety of additional metrics). Further, Fig. C2 extends Fig. 8 and shows empirical CDFs of the experiment with regional forcings for additional metrics. We refer to Gauch et al. (2023) for the definitions of these measures.

Lastly, Fig. C3 shows the fractional attention to each forcing product for three models trained with different random seeds (see experiment 1).

Figure C1. Extended version of Fig. 6, showing additional metrics (see Gauch et al. (2023) for the definitions of these metrics).

Figure C2. Extended version of Fig. 8, showing additional metrics (see Gauch et al. (2023) for the definitions of these metrics).

**Figure C3.** Fraction of attention each product received at each basin, averaged over time. The pie slices are scaled by their fraction to (overly) emphasize differences. Each subplot shows the results for a model trained with a different seed. For better overview, we only plot a random sample of 100 gauges.

**Figure D1.** High-level illustration of attention. The query vector (left) is compared to each key vector (middle), and the corresponding value vectors are merged in a weighted average according to the similarity measure, producing the attention output (right).

# Appendix D: A very brief introduction to attention

275

This section gives a brief high-level introduction to attention, since, as of now, attention is not a widely used concept in hydrologic deep learning applications. As the name suggests, the main idea of "attention" is to provide neural networks with a way to focus on specific parts of their inputs, depending on the current context. Early attention mechanisms come from language applications (Graves, 2013; Bahdanau et al., 2015), where models would focus on relevant words in the source language to produce the corresponding translated words in the target language. With the introduction of the Transformer architecture, attention became one of the most widely used concepts in deep learning (Vaswani et al., 2017). By now, attention and similar approaches have made their way into applications in various fields, including hydrology (e.g., Auer et al., 2024; Rasiya Koya and Roy, 2024).

One way to think about attention—and the origin of today's query/key/value nomenclature—is as a learned similarity-based soft database retrieval (Fig. D1). Let us deconstruct this: by "database", we refer to pairs of so-called *values* and *keys*. That is, each value is an entry in the database that we can retrieve with its associated key. Given a *query*, we calculate a similarity score between the query and each key (this constitutes the "similarity-based" component). All three elements (query/keys/values) are network embeddings, i.e., vectors. For example, one could embed a timeseries of runoff observations as keys, create a one-to-one mapping to the values and then use a given event as the query to search for similar occurrences. The output of the attention operation is a weighted mean of all values, where the weight is higher for values whose keys are more similar to the query (hence "soft" lookup; we do not return a specific value from the database but a weighted average across all values). For example, if we use attention for a translation task, the query would be a learned embedding of the word currently being processed, and keys and values would be embeddings of all source language words.<sup>3</sup> By adjusting the embedding networks, the model can now learn to achieve higher similarity between query and words that are relevant for translating the current

<sup>&</sup>lt;sup>3</sup>We ignore some specifics to language modeling here (e.g., positional encoding or tokenization), because they are not immediately relevant to the attention mechanism at the high level of our explanation.

word and lower similarity between query and irrelevant words. Finally, we can apply masking (setting the similarity to zero) to disallow attention to certain words.

While the most common application of attention is retrieval along a temporal axis (such as the progression of a sentence), the concept generalizes to retrieval of values from arbitrary sets (Dosovitskiy et al., 2021; Ramsauer et al., 2021). In this paper, we consider the embeddings of meteorological forcings as our key–value database (the embeddings act both as keys and as values), and the static attributes of a basin as our query. Hence, the model can learn to retrieve different forcing combinations in different places.

We conclude this short introduction with the caveat that deep learning is an active field, and at this point there are thousands of publications leveraging, improving, or analyzing attention mechanisms. Therefore, this introduction is by far not exhaustive, nor does it cover any of the formal and mathematical aspects. For a deeper introduction, including the actual equations, we refer to Alammar (2018); Rohrer (2021); Bishop and Bishop (2023).

290 *Author contributions.* MG, FK, and DK developed the idea, conceptualization, and methods of the paper. MG wrote the code and ran the experiments. All authors were involved in the writing of the paper.

Competing interests. One author (DK) is member of the editorial board of HESS.

280

285

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
