# Peer review of "How to deal w\_\_\_ missing input data"

_EGUsphere, 2025_

## Referee Comment (RC1)

**Review**

Gauch, M., Kratzert, F., Klotz, D., Nearing, G., Cohen, D., and Gilon, O.:
How to deal w___ missing input data
EGUsphere [preprint]
https://doi.org/10.5194/egusphere-2025-1224, 2025.

Dear Martin and others,

It was a pleasure to review your manuscript on "*How to deal w___ missing input data*", submitted to the Hydrology and Earth System Science journal. I found your study to be well-structured, informative, and a very pleasant read. The experiments are clearly motivated, performed, and analyzed. I think this is a valuable contribution to the field.

I have provided a list of minor comments for your consideration below. Most of these do not require urgent revisions. Most comments are regarding clarity especially for readers that may not have much experience regarding methods like embedding and attention. The authors have already provided additional material which is much appreciated.

I have no doubts that the authors will be able to respond to all my comments without any problem. Hence, I am recommending minor revisions. I appreciate the effort you have put into this work and would be happy to have another look at the revised version.

Best regards,
Julie Mai

Detailed comments:

***Introduction***
- Title & Abstract: I like the title and the clue that this manuscript deals with missing information. Maybe it would be good to make that connection at the end (or elsewhere) in the abstract. Something like: "[…] or not arrive at all; like the missing characters in 'to deal w___ missing input data'."
- Figure 1: Mention in the caption that "gray" indicates missing data and shades of "blue" mean data available (?!). What I don't understand is why the yellow box indicates where models are not robust. Isn't that where all data across both basins are available and complete and hence this is where the model is robust (given that only forcing group 1 and 2 are used and 3 is discarded)? Some more detailed explanation may be required.

- Lines 37-42: Would it be possible to make a connection of these three cases to Figure 1? E.g., first setting is top case in Fig 1? Maybe there is a way to adjust Figure 1 to be used not only for illustrating various kinds of missing data but also used for these three cases? Optional to address. May just be helpful for readers to understand the three experiments better.
- Lines 52-53: Use of term "downstream model". I am assuming that it is "downstream" in the modeling process and not downstream in a hydrologic sense. It may be good to clarify this.

**Data and methods**
- Footnote on page 3: "Unlike what is mentioned in the paper, …". I am assuming that the Kratzert et al. (2021) is referred to. I haven't consulted both tables of static attributes yet and am confused if "p_seasonality" is or is not a static attribute in the study presented here. Maybe this footnote would be better placed with the list of actual static attributes to be less confusing as I assume that using/not using this static attribute does not make a huge difference?!
- Figure 2:
    - "NaNs in the input data for a given time step are replaced by zeros, …" —> I would mention that this is what happened to the three example entries for forcing group 2 (grey means zero).
    - I would recommend having four instead of three example entries for each forcing group. This way it becomes clearer that the binary flags are per forcing group and not per basin.
    - Maybe mention that this example shows forcing available for three/four basins. Initially I thought time steps… I know it is all in the caption, but it took me a second to wrap my head properly around this.
    - Why is the third entry in forcing group 3 not set to zero even though it seems it is not available (NaN) for one of the basins? I would recommend explaining this a bit more in detail. This seems crucial as the binary flag seems to be only set to 0 if a forcing is not available at any basin.
- Lines 82-83: "each of them yielding an embedding vector of the same size" —> Not an expert of embedding networks. What is the size of the resulting embedding vector. In your example (Fig 3) it is 4. Is that a hyper-parameter of the embedding network? Some information on that may be helpful.
- Line 83: "average the non-NaN embeddings" —> Again, my limited knowledge of embeddings here: Under which circumstances can embeddings be NaN? It may be helpful to have such an example in your Figure 3 (vectors shown as list in "avg()").
- Lines 92-93: "Appendix D provides a brief introduction to the concept of attention for readers who are not familiar with the topic." —> Much appreciated! Great job explaining this with intuitive examples.
- Figure 4:
    - There is some overlap of text in top left corner.
    - Is it a coincidence that there are 4 static attributes and the length of your embeddings is 4 as well?
    - What are the binary flags used for? Aren't they already implicit by you only creating two embeddings and none for forcing group 2? If the binary flags are important to inform the

number of embeddings created, why are the binary flags not part of the masked mean figure (Fig 3)? Guessing that this is a "requirement" of the attention framework?!
  - Make explicit in caption that "k/v" means "keys and values".
- Lines 96-98: In the masked mean you decided not to use static attributes as they only deteriorated the performance. For attention networks you decided to use them. I am assuming that the performance increased by using them. If so, I would mention this like you reporting on this already for the masked mean (lines 84-86).
- Figure 5: The caption does not make a connection to the legend. I know the text explains that the (random) subset of basins was assumed to have three data products while the rest is assumed to have only 2 products available, but this should be put in the caption to explain the figure without the need to look up in the text what it actually means.
- Experiment 3: Lines 129-137: What do you think is the impact of you picking the worst performing of the three products, i.e., NLDAS (see lines 115-116) as the additional one for the subset of 51 basins? Your motivation for this experiment was that "higher quality" forcing may be available locally. Shouldn't you have picked the best performing forcing as the additional one?

**Results**
- Figure 6:
  - "NaN probability" equals "p_time", right?
  - I would probably sort the various approaches in the legend in the order they are introduced (replacing, masked mean, and attention). Same for figures 7 and 8.
  - I'd potentially cite Kratzert et al. (2021) for the two reference results (dashed and dotted line) to emphasize that this was done previously. Again, this is only to make the figure content somewhat independent from the rest of the text.
- Line 145: "while masked mean is slightly better in KGE" —> it looks like input replacing is also better in KGE at p_time=0.0.
- Lines 149-150: "except for p_time = 0.2, the masked mean results are significantly better than those of input replacing" —> I can't really see that in the plot. The median value for blue (masked mean) and yellow (input replacing) seem both to be around 0.775. Are they only significantly different in terms of the one-sides Wilcoxon signed-rank test? Is that the only result where a significant difference was detected?
- Line 161: "exact forcings that are available at inference time" —> "exact forcings that are available at inference time (dashed line)"
- Line 161: "performs significantly better" —> based on a statistical test or just from looking at plots? I can't really see a difference between the dashed CDF lines and the colored CDF lines; especially in the right column of Figure 7. It may be that you are talking in this paragraph about the single forcing results (left column) but it's not clear from the text. Only when you start the next paragraph it is suggesting that you were only looking at single-set results (probably)?
- Line 160: Is the highlighted result for NLDAS-only likely happening because NLDAS was the worst performing dataset in general?

- Line 172: "The three-forcing model trained only on the 51 basins ..." —> Maybe call this "regional model" here explicitly or say that this refers to the dashed line in the plot to make it easier to connect these things
- Figure 8: Can "regional model" in the legend include that it is using all three forcings? Like the legend entry for the "global model". Easier to connect to the text (line 172 etc.).
- Lines 174-175: "However, from a practical hydrological perspective, all approaches perform quite similar, despite the statistical significance." —> I like this sentence. I think this may also be something that could be added to the results for experiment 2.

**Discussion and conclusions**
- Line 183: "unable to outperform the baseline trained on all three forcings but only 51 local basins (experiment 3)" —> may this also be caused by the additional third forcing not being "better" than the others but indeed shown previously to be the least performing?
- General: Is there any notable difference in computational expense or difficulty in implementation between the three methods? This may also be factors for practitioners to select one method over another.

**Appendix**
- Figure C1/C2: It would be great if you could add a reference for the definition of all the additional metrics.

---

## Author Comment (AC2)

**Responses to Reviews**
**How to deal w___ missing input data**

https://doi.org/10.5194/egusphere-2025-1224

**RC1**

Dear Martin and others,

It was a pleasure to review your manuscript on "*How to deal w___ missing input data*", submitted to the Hydrology and Earth System Science journal. I found your study to be well-structured, informative, and a very pleasant read. The experiments are clearly motivated, performed, and analyzed. I think this is a valuable contribution to the field.

I have provided a list of minor comments for your consideration below. Most of these do not require urgent revisions. Most comments are regarding clarity especially for readers that may not have much experience regarding methods like embedding and attention. The authors have already provided additional material which is much appreciated.

I have no doubts that the authors will be able to respond to all my comments without any problem. Hence, I am recommending minor revisions. I appreciate the effort you have put into this work and would be happy to have another look at the revised version.

Best regards,
Julie Mai

*Thank you Julie for your very detailed review! Please see our inline answers below.*

Detailed comments:

***Introduction***
- Title & Abstract: I like the title and the clue that this manuscript deals with missing information. Maybe it would be good to make that connection at the end (or elsewhere) in the

abstract. Something like: "[…] or not arrive at all; like the missing characters in 'to deal w___ missing input data'."

Explaining a joke ruins it, so we prefer to get the smiles of those who get it and let everyone else discuss the paper with their colleagues to understand.

- Figure 1: Mention in the caption that "gray" indicates missing data and shades of "blue" mean data available (?!). What I don't understand is why the yellow box indicates where models are not robust. Isn't that where all data across both basins are available and complete and hence this is where the model is robust (given that only forcing group 1 and 2 are used and 3 is discarded)? Some more detailed explanation may be required.

We've clarified that gray bars are missing data, thanks, that is a good point. As for the yellow box: The annotation "model not robust" here refers to a model that has no mechanism to work with missing data. Such a model could only be trained on the subset of data represented by the yellow box. I.e., it could not make use of forcing group 3 and could only use less than half of the time (represented by the horizontal axis in that figure). The opposite is true for a model that is robust against missing data (e.g., using any of the methods presented in this manuscript). Such a model could use all of the available data inside of the purple box. We have slightly expanded the caption to hopefully make this point more clear.

- Lines 37-42: Would it be possible to make a connection of these three cases to Figure 1? E.g., first setting is top case in Fig 1? Maybe there is a way to adjust Figure 1 to be used not only for illustrating various kinds of missing data but also used for these three cases? Optional to address. May just be helpful for readers to understand the three experiments better.

Good idea, done.

- Lines 52-53: Use of term "downstream model". I am assuming that it is "downstream" in the modeling process and not downstream in a hydrologic sense. It may be good to clarify this.

The word "downstream" was not necessary in the sentence that already says "subsequently". Deleted.

**Data and methods**
- Footnote on page 3: "Unlike what is mentioned in the paper, …". I am assuming that the Kratzert et al. (2021) is referred to. I haven't consulted both tables of static attributes yet and am confused if "p_seasonality" is or is not a static attribute in the study presented here. Maybe this footnote would be better placed with the list of actual static attributes to be less confusing as I assume that using/not using this static attribute does not make a huge difference?!

Yes, we were referring to Kratzert et al. We've updated the footnote to clarify that. And as you say, we don't expect this to make much of a difference.

- Figure 2:

- "NaNs in the input data for a given time step are replaced by zeros, …" —> I would mention that this is what happened to the three example entries for forcing group 2 (grey means zero).
  Done.

- I would recommend having four instead of three example entries for each forcing group. This way it becomes clearer that the binary flags are per forcing group and not per basin.
  Unfortunately, it's complicated. If we use four boxes in the forcing groups, the embeddings end up having the same size as the inputs which is also not ideal (we'd like to emphasize that the embedding size can be entirely different from the number of inputs). So to disambiguate all of these cases, we'd end up with a lot of boxes and cluttered illustrations. We have instead expanded the caption of the figure (see next response) to further clarify what each visual element represents.

- Maybe mention that this example shows forcing available for three/four basins. Initially I thought time steps… I know it is all in the caption, but it took me a second to wrap my head properly around this.
  The boxes of each forcing group are intended to represent different variables (like precipitation, temperature, …) and not basins. That is why different products have different amounts of boxes. We can see how that was not clear from the figure. Unfortunately, the more boxes we add the more complicated (and the harder to layout) the figures become. Instead, we have added a brief explanation in the caption.

- Why is the third entry in forcing group 3 not set to zero even though it seems it is not available (NaN) for one of the basins? I would recommend explaining this a bit more in detail. This seems crucial as the binary flag seems to be only set to 0 if a forcing is not available at any basin.
  See above (boxes are variables, not basins). We have changed the description to clarify.

- Lines 82-83: "each of them yielding an embedding vector of the same size" —> Not an expert of embedding networks. What is the size of the resulting embedding vector. In your example (Fig 3) it is 4. Is that a hyper-parameter of the embedding network? Some information on that may be helpful.
  Yes, this is a hyperparameter. Appendix A describes this in more detail — we tuned both the hidden and output size of the embedding networks.

- Line 83: "average the non-NaN embeddings" —> Again, my limited knowledge of embeddings here: Under which circumstances can embeddings be NaN? It may be helpful to have such an example in your Figure 3 (vectors shown as list in "avg()").
  The NaN embeddings are exactly the ones that correspond to providers which were

unavailable at a given time step. We've clarified this in the updated manuscript.

- Lines 92-93: "Appendix D provides a brief introduction to the concept of attention for readers who are not familiar with the topic." —> Much appreciated! Great job explaining this with intuitive examples.
Thanks!

- Figure 4:
    - There is some overlap of text in top left corner.
    We've added a bit more space between "static attributes" and "binary flags" to better separate those two labels visually.

    - Is it a coincidence that there are 4 static attributes and the length of your embeddings is 4 as well?
    Yes. In practice there are 26 static attributes and the embeddings have a size of 10.

    - What are the binary flags used for? Aren't they already implicit by you only creating two embeddings and none for forcing group 2? If the binary flags are important to inform the number of embeddings created, why are the binary flags not part of the masked mean figure (Fig 3)? Guessing that this is a "requirement" of the attention framework?!
    The binary flags in the attention allow the model to explicitly change the query depending on feature availability. It is true that implicitly the model also knows about feature availability without those flags, but it might be harder to use that information if it is only available implicitly through the unavailability of certain key/value vectors.

    - Make explicit in caption that "k/v" means "keys and values".
    Done.

- Lines 96-98: In the masked mean you decided not to use static attributes as they only deteriorated the performance. For attention networks you decided to use them. I am assuming that the performance increased by using them. If so, I would mention this like you reporting on this already for the masked mean (lines 84-86).
The reason we use static attributes for the attention mechanism is that we need some sort of input that forms the query. Only using the binary flags as a query would make it impossible for the model to shift its attention depending on the location, which was the premise of the architecture. Hence, we did not explore attention without the static inputs. We expanded the description of the attention approach to further explain the purpose of the static attributes.

- Figure 5: The caption does not make a connection to the legend. I know the text explains that the (random) subset of basins was assumed to have three data products while the rest is assumed to have only 2 products available, but this should be put in the caption to explain the

figure without the need to look up in the text what it actually means.

Adapted the caption to be more verbose.

- Experiment 3: Lines 129-137: What do you think is the impact of you picking the worst performing of the three products, i.e., NLDAS (see lines 115-116) as the additional one for the subset of 51 basins? Your motivation for this experiment was that "higher quality" forcing may be available locally. Shouldn't you have picked the best performing forcing as the additional one?

Interestingly, despite NLDAS being the worst single-forcing model, Figure 3 from Kratzert et al. (2021) shows that the largest gap between two- and three-forcing models occurs when holding out NLDAS. An explanation for this behavior might be that NLDAS could represent the "odd one out", i.e., even though being wrong more often, its errors might have little correlation with those of Daymet and Maurer and therefore present additional useful information.

**Results**
- Figure 6:
  - "NaN probability" equals "p_time", right?

    Yes.

  - I would probably sort the various approaches in the legend in the order they are introduced (replacing, masked mean, and attention). Same for figures 7 and 8.

    Done.

  - I'd potentially cite Kratzert et al. (2021) for the two reference results (dashed and dotted line) to emphasize that this was done previously. Again, this is only to make the figure content somewhat independent from the rest of the text.

    Done.

- Line 145: "while masked mean is slightly better in KGE" —> it looks like input replacing is also better in KGE at p_time=0.0.

  Correct, updated the description to reflect that.

- Lines 149-150: "except for p_time = 0.2, the masked mean results are significantly better than those of input replacing" —> I can't really see that in the plot. The median value for blue (masked mean) and yellow (input replacing) seem both to be around 0.775. Are they only significantly different in terms of the one-sides Wilcoxon signed-rank test? Is that the only result where a significant difference was detected?

  Our statement is correct: p_time = 0.2 is the only setting where the one-sided Wilcoxon test cannot reject the null hypothesis "masked mean performs equally or less good than input replacing" at α = 0.05. Note that the results of the (paired) statistical tests do not necessarily

have to be clear from the figure that shows only the medians. In a different figure (e.g., a CDF comparing the results of just p_time=0.2), it might visually be more clear. Here, however, we want to show the performance as p_time increases, which is hard to also combine with the full distribution of values at each p_time value without cluttering the figure. Nevertheless, visually, the figure is in agreement with those results: p_time = 0.2 is the only point along the x-axis where the yellow shading (input replacing) runs higher than the blue shading (masked mean).

- Line 161: "exact forcings that are available at inference time" —> "exact forcings that are available at inference time (dashed line)"
  Done.

- Line 161: "performs significantly better" —> based on a statistical test or just from looking at plots? I can't really see a difference between the dashed CDF lines and the colored CDF lines; especially in the right column of Figure 7. It may be that you are talking in this paragraph about the single forcing results (left column) but it's not clear from the text. Only when you start the next paragraph it is suggesting that you were only looking at single-set results (probably)?
  Based on a statistical test (we made sure to only use the word "significant" in the scientific sense).
  You are right that the wording was confusing; we are indeed focusing on the single-forcing experiments (left column of Fig. 7) in this paragraph. We've clarified this in the updated manuscript.

- Line 160: Is the highlighted result for NLDAS-only likely happening because NLDAS was the worst performing dataset in general?
  While we cannot say for sure, this seems like a plausible explanation. Note, however, that the effect sizes are very small in all cases, so the practical implications of this finding seem limited to us.

- Line 172: "The three-forcing model trained only on the 51 basins …" —> Maybe call this "regional model" here explicitly or say that this refers to the dashed line in the plot to make it easier to connect these things
  Good point, changed.

- Figure 8: Can "regional model" in the legend include that it is using all three forcings? Like the legend entry for the "global model". Easier to connect to the text (line 172 etc.).
  Done.

- Lines 174-175: "However, from a practical hydrological perspective, all approaches perform quite similar, despite the statistical significance." —> I like this sentence. I think this may also

be something that could be added to the results for experiment 2.
Added similar wording to experiment 2.

**Discussion and conclusions**
- Line 183: "unable to outperform the baseline trained on all three forcings but only 51 local basins (experiment 3)" —> may this also be caused by the additional third forcing not being "better" than the others but indeed shown previously to be the least performing?
See our response above; holding out NLDAS is expected to cause the largest deterioration (at least on the CONUS-wide comparison from Kratzert et al. (2021). Nevertheless, at a higher level, we agree with the  reasoning behind your argument, as we already state the high-quality forcings a likely reason for the limited differences between approaches.

- General: Is there any notable difference in computational expense or difficulty in implementation between the three methods? This may also be factors for practitioners to select one method over another.
The attention mechanism is certainly more tricky to implement than the other two approaches, which contributes to our conclusion that "Therefore, in its current form, attention appears unnecessary" (L196). The other two approaches are quite simple to implement and (compared to the core LSTM time series model component) have minor effects on computational expense, so we would not consider this a major factor in the decision for one of the approaches.

**Appendix**
- Figure C1/C2: It would be great if you could add a reference for the definition of all the additional metrics.
Done.

---

## Author Comment (AC3)

**Responses to Reviews**
**How to deal w___ missing input data**

https://doi.org/10.5194/egusphere-2025-1224

**RC2**

**General comments**

The paper "How to deal w___ missing input data" provides a thorough implementation and analysis of three strategies for how to deal with missing input data for operational AI-based models that depend on the real-time availability of meteorological forcings. The strategies are: input replacing, masked mean, and attention. Through three sets of experiments that train models with different permutations of missing forcings, they show that the masked mean strategy tends to perform the best, if only marginally. They show that attention appears to be unnecessary, as it is complicated and tends to collapse to the simpler weighted mean approach. This is a very nice result.

While the paper is well structured, the text requires some refinement. The description of the key experiments lack some details and make it difficult for the reader to follow. For instance, experiment 2 can only be understood after reading the figure caption, not from the text body. Furthermore, the reviewer has some concerns regarding the reproducibility, since the training code is not made available. Combined with some technical errors, a major revision is recommended.

We'd like to thank the reviewer for their detailed comments. We have responded to each comment directly below.

**Specific comments**

**Major concerns**

- Code is missing. While the authors provide detailed code for the replication of the figures as well as the trained models, the code for the model training and inference is not available. In my opinion, and in line with GMD's practices, the code should be made available before this manuscript can be published.
This is not correct. The code to reproduce our experiments is publicly available as part of the NeuralHydrology Python library as it is already described in the code & data availability section of the paper.

- The description of the LSTM model is missing in this manuscript. While the authors provide a thorough explanation of the model architectures for the input-data-processing

methods, a description of the architecture for the core LSTM model is missing. The description of the target output of the LSTM model (i.e., what it is actually predicting) is also missing. i.e., Section 2.1 describes the forcing variables, but it is unclear what the target is.

You are right that we missed to explicitly state the target variable (streamflow); we have corrected that in the updated manuscript. We have further added a brief statement to explain that, apart from the input layers, our model architecture is identical to that of (e.g.) Kratzert et al. (2020). We believe that it is not necessary to reiterate the LSTM architecture in detail, as there are no changes to what has been described in much of the cited prior work.

- Section 2.1: Is there a mistake in the testing period? The testing period should not overlap with the training dataset; otherwise, this is a major error, as the models could be overfitting.

This was a typo in the text, thank you for catching it. The correct periods are: 1 October 1999 to 30 September 2008 for training, 1 October 1980 to 30 September 1989 for validation, and 1 October 1989 to 30 September 1999 for testing. Hence, there is no data leakage between training and validation/test periods.

- Starting from section 3.2, following and understanding the experiment setup is difficult and needs the be clarified. For instance, in section 3.2, you mention that the results show the cases of missing forcings at inference. For a given line/plot, it is not clear what parts of each model are trained (just the LSTM, or also the data-preprocessing components?). It is not also clear what the difference is between the dashed lines and the masked mean/attention/input replacement lines are. I understood the dashed line as a LSTM model trained with 1 (or 2) forcings, and the solid lines are the same LSTM, but with an additional trained encoder (with the appropriate data-processing techniques). However, I could be misunderstanding this. For the dashed (reference) lines, how are the forcings passed to the LSTM (concat, mean, attention?).

We trained the entire model from scratch in all experiments. We are not sure where the idea that only parts of the model might be trained originates from.

The dashed and dotted lines stem from Kratzert et al., 2021, i.e., from models that are not robust to missing input data. For these models, the forcings are simply concatenated, following the standard training approach for LSTMs. We've updated the description of experiment 2 and rephrased the caption of Figure 7 to better guide readers and help to avoid confusion.

Minor concerns and questions

- Figure 6: When you describe the attention mechanism for high probability p, it is an interesting result that the weights fluctuate around 1/3. Is this also true for the lower missing data probabilities, where the attention method performs worse? If not, what would your analysis be around this?

Our finding about the equal attention weights was actually run on a NaN probability of zero. We can see that the previous sentence about high NaN probability suggested otherwise, so we have updated the text to state this clearly. Further, in preparation for this response, we re-ran the analysis with p_time = 0.6 and found the effect to be similar there.

- From reading the manuscript, it is unclear to the reader why the metric becomes CDF for experiments 2 and 3. Please highlight the connection between these two metrics in the text, and specify that the ideal result is a delta function at NSE=1, and thus, curves that are closer to the right are better.
  The main metrics (in the main paper) is NSE and/or KGE in all experiments (the appendix shows additional metrics). We chose cumulative density functions (CDFs) to visualize the distribution of metric values in two of the experiments. As experiment 1 further shows the uncertainty across multiple seeds, we felt that a CDF would be too cluttered and hard to read.
  CDFs are extremely common in hydrologic literature, so we do not see a need to explicitly describe how to read them. Nevertheless, we added the explanation that curves further to the right are better.

**Specific, minor comments**

- The abstract is missing key results. It would be beneficial to briefly mention the different solutions and hint at which provide the most robust results (seemingly, masked mean)
  We agree and have expanded the abstract in the revised manuscript to include short descriptions of the methods and a peek at the results.

- Minor comment: for readability, it would be a smoother flow if the literature review section on other fields and models with missing input data occurs earlier in the introduction, before you present the three strategies to accomplish this goal.
  We would like to push back on this, as we believe that it makes sense to explain early in the manuscript how we intend to achieve the goals outlined at the beginning of the paper. This gives a high-level overview to the reader, before we dive into the details of related approaches. It also allows us to point out differences between those approaches and our proposed methods that would be hard to understand if we hadn't introduced our methods beforehand.

---

## Author Response (AR2)

**Authors' Response**

Again, we would like to thank the reviewers and the editor for their feedback.

Regarding the comment about overlapping text in the figure: we realized that this seems to happen only on certain operating systems/pdf viewers (specifically it seems to happen on Macs). We have fixed this by changing the figure to an image format.